# WHEN SHOULD AGENTS EXPLORE?

**Miruna Pîslar**    **David Szepesvari**    **Georg Ostrovski**    **Diana Borsa**    **Tom Schaul**

DeepMind London, UK
{mirunapislar,dsz,ostrovski,borsa,tom}@deepmind.com

## ABSTRACT

Exploration remains a central challenge for reinforcement learning (RL). Virtually all existing methods share the feature of a *monolithic* behaviour policy that changes only gradually (at best). In contrast, the exploratory behaviours of animals and humans exhibit a rich diversity, namely including forms of *switching* between modes. This paper presents an initial study of mode-switching, non-monolithic exploration for RL. We investigate different modes to switch between, at what timescales it makes sense to switch, and what signals make for good switching triggers. We also propose practical algorithmic components that make the switching mechanism adaptive and robust, which enables flexibility without an accompanying hyper-parameter-tuning burden. Finally, we report a promising and detailed analysis on Atari, using two-mode exploration and switching at sub-episodic time-scales.

## 1 INTRODUCTION

The trade-off between exploration and exploitation is described as the crux of learning and behaviour across many domains, not just reinforcement learning (Sutton & Barto, 2018), but also in decision making (Cohen et al., 2007), evolutionary biology (Cremer et al., 2019), ecology (Kembro et al., 2019), neuroscience (e.g., focused versus diffuse search in visual attention (Wolfe et al., 1989), dopamine regulations (Chakroun et al., 2020)), cognitive sciences (Hills et al., 2015), as well as psychology and psychiatry (Addicott et al., 2017). In a nutshell, exploration is about the balance between taking the familiar choice that is known to be rewarding and learning about unfamiliar options of uncertain reward, but which could ultimately be more valuable than the familiar options.

Ample literature has studied the question of *how much* to explore, that is how to set the overall trade-off (and how to adjust it over the course of learning) (Jaksch et al., 2010; Cappé et al., 2013; Lattimore & Szepesvári, 2020; Thrun, 1992), and the question of *how* to explore, namely how to choose exploratory actions (e.g., randomly, optimistically, intrinsically motivated, or otherwise) (Schmidhuber, 1991; Oudeyer & Kaplan, 2009; Linke et al., 2019). In contrast, the question of *when* to explore has been studied very little, possibly because it does not arise in bandit problems, where a lot of exploration methods are rooted. The 'when' question and its multiple facets are the subjects of this paper. We believe that addressing it could lead to more *intentional* forms of exploration.

Consider an agent that has access to two *modes* of behaviour, an 'explore' mode and an 'exploit' mode (e.g., a random policy and a greedy policy, as in $\varepsilon$-greedy). Even when assuming that the overall proportion of exploratory steps is fixed, the agent still has multiple degrees of freedom: it can explore more at the beginning of training and less in later phases; it may take single exploratory steps or execute prolonged periods of exploration; it may prefer exploratory steps early or late within an episode; and it could trigger the onset (or end) of an exploratory period based on various criteria. Animals and humans exhibit non-trivial behaviour in all of these dimensions, presumably encoding useful inductive biases that way (Power, 1999). Humans make use of multiple effective strategies, such as selectively exploring options with high uncertainty (a form of directed, or information-seeking exploration), and increasing the randomness of their choices when they are more uncertain (Gershman, 2018; Gershman & Tzovaras, 2018; Ebitz et al., 2019). Monkeys use directed exploration to manage explore-exploit trade-offs, and these signals are coded in motivational brain regions (Costa et al., 2019). Patients with schizophrenia register changes in directed exploration and expe-

rience low-grade inflammation when shifting from exploitation to random exploration (Waltz et al., 2020; Cathomas et al., 2021). This diversity is what motivates us to study which of these can benefit RL agents in turn, by expanding the class of exploratory behaviours beyond the commonly used *monolithic* ones (where modes are merged homogeneously in time).

## 2    METHODS

The objective of an RL agent is to learn a policy that maximises external reward. At the high level, it achieves this by interleaving two processes: generating new experience by interacting with the environment using a behaviour policy (exploration) and updating its policy using this experience (learning). As RL is applied to increasingly ambitious tasks, the challenge for exploration becomes to keep producing *diverse* experience, because if something has not been encountered, it cannot be learned. Our central argument is therefore simple: a monolithic, time-homogeneous behaviour policy is strictly less diverse than a heterogeneous mode-switching one, and the former may hamstring the agent's performance. As an illustrative example, consider a human learning how to ride a bike (explore), while maintaining their usual happiness through food, sleep, work (exploit): there is a stark contrast between a monolithic, time-homogeneous behaviour that interleaves a twist of the handlebar or a turn of a pedal once every few minutes or so, and the mode-switching behaviour that dedicates prolonged periods of time exclusively to acquiring the new skill of cycling.

While the choice of behaviour in pure exploit mode is straightforward, namely the greedy pursuit of external reward (or best guess thereof), denoted by $\mathcal{G}$, there are numerous viable choices for behaviour in a pure explore mode (denoted by $\mathcal{X}$). In this paper we consider two standard ones: $\mathcal{X}_U$, the naive uniform random policy, and $\mathcal{X}_I$, an intrinsically motivated behaviour that exclusively pursues a novelty measure based on random network distillation (RND, (Burda et al., 2018)). See Section 4 and Appendix B for additional possibilities of $\mathcal{X}$. In this paper we choose fixed behaviours for these modes, and focus solely on the question of *when* to switch between them. In our setting, overall proportion of exploratory steps (the *how much*), denoted by $p_{\mathcal{X}}$, is not directly controlled but derives from the *when*.

### 2.1    GRANULARITY

An exploration *period* is an uninterrupted sequence of steps in explore mode. We consider four choices of temporal granularity for exploratory periods, also illustrated in Figure 1.

**Step-level** exploration is the most common scenario, where the decision to explore is taken independently at each step, affecting one action.[1] The canonical example is $\varepsilon$-greedy (Fig.1:C).

**Experiment-level** exploration is the other extreme, where all behaviour during training is produced in explore mode, and learning is off-policy (the greedy policy is only used for evaluation). This scenario is also very common, with most forms of intrinsic motivation falling into this category, namely pursuing reward with an intrinsic bonus throughout training (Fig.1:A).[2]

**Episode-level** exploration is the case where the mode is fixed for an entire episode at a time (e.g., training games versus tournament matches in a sport), see Fig.1:B. This has been investigated for simple cases, where the policy's level of stochasticity is sampled at the beginning of each episode (Horgan et al., 2018; Kapturowski et al., 2019; Zha et al., 2021).

**Intra-episodic** exploration is what falls in-between step- and episode-level exploration, where exploration periods last for multiple steps, but less than a full episode. This is the least commonly studied scenario, and will form the bulk of our investigations (Fig.1:D,E,F,G).

We denote the length of an exploratory period by $n_{\mathcal{X}}$ (and similarly $n_{\mathcal{G}}$ for exploit mode). To characterise granularity, our summary statistic of choice is $\mathrm{med}_{\mathcal{X}} := \mathrm{median}(n_{\mathcal{X}})$. Note that there are two possible units for these statistics: the raw steps or the proportion of the episode length $L$.

---

[1]The length of an exploratory period tends to be short, but it can be greater than 1, as multiple consecutive step-wise decisions to explore can create longer periods.

[2]Note that it is also possible to interpret $\varepsilon$-greedy as experiment-level exploration, where the $\mathcal{X}$ policy is fixed to a noisy version of $\mathcal{G}$.

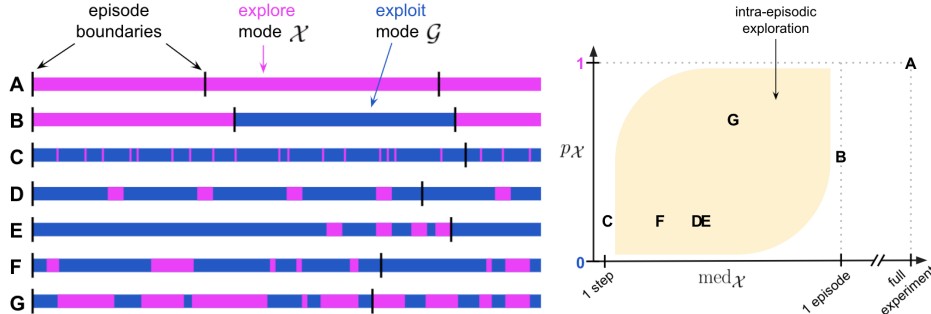

Figure 1: Illustration of different types of temporal structure for two-mode exploration. **Left**: Each line A-G depicts an excerpt of an experiment (black lines show episode boundaries, experiment continues on the right), with colour denoting the active mode (blue is exploit, magenta is explore). **A** is of experiment-level granularity, **B** episode-level, **C** step-level, and **D-G** are of intra-episodic exploration granularity. **Right**: The same examples, mapped onto a characteristic plot of summary statistics: overall exploratory proportion $p_\mathcal{X}$ versus typical length of an exploratory period $\mathrm{med}_\mathcal{X}$. The yellow-shaded area highlights the intra-episodic part of space studied in this paper (some points are not realisable, e.g., when $p_\mathcal{X} \approx 1$ then $\mathrm{med}_\mathcal{X}$ must be large). **C, D, E, F** share the same $p_\mathcal{X} \approx 0.2$, while interleaving exploration modes in different ways. **D** and **E** share the same $\mathrm{med}_\mathcal{X}$ value, and differ only on whether exploration periods are spread out, or happen toward the end.

The latter has different (relative) semantics, but may be more appropriate when episode lengths vary widely across training. We denote it as $\mathrm{rmed}_\mathcal{X} := \mathrm{median}(n_\mathcal{X}/L)$.

## 2.2 SWITCHING MECHANISMS

Granularity is but the coarsest facet of the 'when' question, but the more precise intra-episode timings matter too, namely when *exactly* to start and when to stop an exploratory period. This section introduces two mechanisms, *blind* and *informed* switching. It is worth highlighting that, in general, the mechanism (or its time resolution) for entering explore mode differs from the one for exiting it (to enter exploit mode) – this asymmetry is crucial to obtain flexible overall amounts of exploration. If switching were *symmetric*, the proportion would be $p_\mathcal{X} \approx 0.5$.

**Blind switching** The simplest switching mechanism does not take any state into account (thus, we call it *blind*) and is only concerned with producing switches at some desired time resolution. It can be implemented deterministically through a counter (e.g., enter explore mode after $100$ exploit mode steps), or probabilistically (e.g., at each step, enter explore mode with probability $0.01$). Its expected duration can be parameterised in terms of raw steps, or in terms of fractional episode length. The opposite of blind switching is *informed* switching, as discussed below.

**Informed switching** Going beyond blind switching opens up another rich set of design choices, with switching informed by the agent's internal state. There are two parts: first, a scalar *trigger* signal is produced by the agent at each step, using its current information – drawing inspiration from human behaviour, we view the triggering signal as a proxy for uncertainty (Schulz et al., 2019): when uncertainty is high, the agent will switch to explore. Second, a binary switching decision is taken based on the trigger signal (for example, by comparing it to a threshold). Again, triggering will generally not be symmetric between entering and exiting an exploratory period.

To keep this paper focused, we will look at one such informed trigger, dubbed 'value promise discrepancy' (see Appendix B for additional competitive variants). This is an online proxy of how much of the reward that the agent's past value estimate promised ($k$ steps ago) have actually come about. The intuition is that in uncertain parts of state space, this discrepancy will generally be larger than when everything goes as expected. Formally,

$$D_{\mathrm{promise}}(t-k, t) := \left| V(s_{t-k}) - \sum_{i=0}^{k-1} \gamma^i R_{t-i} - \gamma^k V(s_t) \right|$$

where $V(s)$ is the agent's value estimate at state $s$, $R$ is the reward, and $\gamma$ is a discount factor.

**Starting mode** When periods last for a significant fraction of episode length, it also matters how the sequence is initialised, i.e., whether an episode starts in explore or in exploit mode, or more generally, whether the agent explores more early in an episode or more later on. It is conceivable that the best choice among these is domain dependent (see Figure 6): in most scenarios, the states at the beginning of an episode have been visited many times, thus starting with exploit mode can be beneficial; in other domains however, early actions may disproportionately determine the available future paths (e.g., build orders in StarCraft (Churchill & Buro, 2011)).

### 2.3 FLEXIBILITY WITHOUT ADDED BURDEN

Our approach introduces additional flexibility to the exploration process, even when holding the specifics of the learning algorithm and the exploration mode fixed. To prevent this from turning into an undue hyper-parameter tuning burden, we recommend adding two additional mechanisms.

**Bandit adaptation** The two main added degrees of freedom in our intra-episodic switching set-up are when (or how often) to enter explore mode, and when (or how quickly) to exit it. These can be parameterised by a duration, termination probability or target rate (see Section 3.1). In either case, we propose to follow Schaul et al. (2019) and Badia et al. (2020a), and delegate the adaptation of these settings to a meta-controller, which is implemented as a non-stationary multi-armed bandit that maximises episodic return. As an added benefit, the 'when' of exploration can now become adaptive to both the task, and the stage of learning.

**Homeostasis** In practice, the scales of the informed trigger signals may vary substantially across domains and across training time. For example, the magnitude of $D_{\mathrm{promise}}$ will depend on reward scales and density and can decrease over time as accuracy improves (the signals could also be noisy). This means that naively setting a threshold hyper-parameter is impractical. For a simple remedy, we have taken inspiration from neuroscience (Turrigiano & Nelson, 2004) to add homeostasis to the binary switching mechanism, which tracks recent values of the signal and adapts the threshold for switching so that a specific average *target rate* is obtained. This functions as an adaptive threshold, making tuning straightforward because the target rate of switching is configured independently of the scales of the trigger signal. See Appendix A for the details of the implementation.

## 3 RESULTS

The design space we propose contains a number of atypical ideas for how to structure exploration. For this reason, we opted to keep the rest of our experimental setup very conventional, and include multiple comparable baselines, ablations and variations.

**Setup: R2D2 on Atari** We conduct our investigations on a subset of games of the Atari Learning Environment (Bellemare et al., 2013), a common benchmark for the study of exploration. All experiments are conducted across 7 games (FROSTBITE, GRAVITAR, H.E.R.O., MONTEZUMA'S REVENGE, MS. PAC-MAN, PHOENIX, STAR GUNNER), the first 5 of which are classified as hard exploration games (Bellemare et al., 2016), using 3 seeds per game. For our agent, we use the R2D2 architecture (Kapturowski et al., 2019), which is a modern, distributed version of DQN (Mnih et al., 2015) that employs a recurrent network to approximate its Q-value function. This is a common basis used in exploration studies (Dabney et al., 2020; Badia et al., 2020b;a). The only major modification to conventional R2D2 is its exploration mechanism, where instead we implement all the variants of mode-switching introduced in Section 2. Separately from the experience collected for learning, we run an evaluator process that assesses the performance of the current greedy policy. This is what we report in all our performance curves (see Appendix A for more details).

**Baselines** There are a few simple baselines worth comparing to, namely the pure explore mode ($p_{\mathcal{X}} = 1$, Fig.1:A) and the pure exploit mode ($p_{\mathcal{X}} = 0$), as well as the step-wise interleaved $\varepsilon$-greedy execution (Fig.1:C), where $p_{\mathcal{X}} = 0.01 = \varepsilon$ (without additional episodic or intra-episodic structure). Given its wide adoption in well-tuned prior work, we expect the latter to perform well overall. The fourth baseline picks a mode for an entire episode at a time (Fig.1:B), with the probability of picking $\mathcal{X}$ being adapted by a bandit meta-controller. We denote these as `experiment-level-X`, `experiment-level-G`, `step-level-0.01` and `episode-level-*` respectively. For each of these, we have a version with uniform ($\mathcal{X}_U$) and intrinsic ($\mathcal{X}_I$) explore mode.

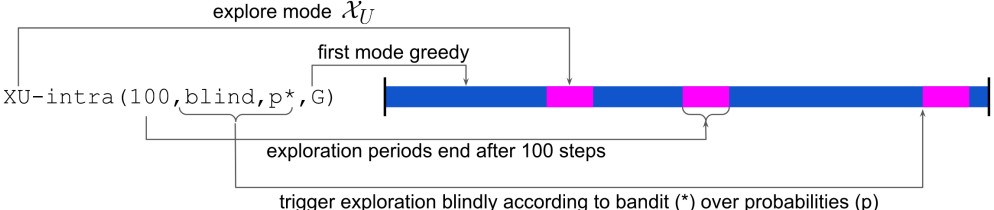

Figure 2: Illustrating the space of design decisions for intra-episodic exploration (see also Figure 9).

## 3.1 VARIANTS OF INTRA-EPISODIC EXPLORATION

As discussed in Section 2, there are multiple dimensions along which two-mode intra-episodic exploration can vary. The concrete ones for our experiments are:

- Explore mode: uniform random $\mathcal{X}_U$, or RND intrinsic reward $\mathcal{X}_I$ (denoted `XU` and `XI`).
- Explore duration ($n_{\mathcal{X}}$): this can be a fixed number of steps $(1, 10, 100)$, or one of these is adaptively picked by a bandit (denoted by `*`), or the switching is symmetric between entering and exiting explore mode (denoted by `=`).
- Trigger type: either `blind` or `informed` (based on value promise, see Section 2.2).
- Exploit duration ($n_{\mathcal{G}}$): for blind triggers, the exploit duration can be parameterised by fixed number of steps $(10, 100, 1000, 10000)$, indirectly defined by a probability of terminating $(0.1, 0.01, 0.001, 0.0001)$, or adaptively picked by a bandit over these choices (denoted by `n*` or `p*`, respectively). For informed triggers, the exploit duration is indirectly parameterised by a target rate in $(0.1, 0.01, 0.001, 0.0001)$, or a bandit over them (`p*`), which is in turn transformed into an adaptive switching threshold by homeostasis (Section 2.2).
- Starting mode: $\mathcal{G}$ greedy (default) or $\mathcal{X}$ explore (denoted by `G` or `X`).

We can concisely refer to a particular instance by a tuple that lists these choices. For example, `XU-intra(100,informed,p*,X)` denotes uniform random exploration $\mathcal{X}_U$, with fixed 100-step explore periods, triggered by the value-promise signal at a bandit-determined rate, and starting in explore mode. See Figure 2 for an illustration.

## 3.2 PERFORMANCE RESULTS

We start by reporting overall performance results, to reassure the reader that our method is viable (and convince them to keep reading the more detailed and qualitative results in the following sections). Figure 3 shows performance across 7 Atari games according to two human-normalised aggregation metrics (mean and median), comparing one form of intra-episodic exploration to all the baselines, separately for each explore mode ($\mathcal{X}_U$ and $\mathcal{X}_I$). The headline result is that intra-episodic exploration improves over both step-level and episode-level baselines (as well as the pure experiment-level modes that we would not expect to be very competitive). The full learning curves per game are found in the appendix, and show scores on hard exploration games like MONTEZUMA'S REVENGE or PHOENIX that are also competitive in absolute terms (at our compute budget of 2B frames).

Note that there is a subtle difference to the learning setups between $\mathcal{X}_U$ and $\mathcal{X}_I$, as the latter requires training a separate head to estimate intrinsic reward values. This is present even in pure exploit mode, where it acts as an auxiliary task only (Jaderberg et al., 2016), hence the differences in pure greedy curves in Figure 3. For details, see Appendix A.

## 3.3 DIVERSITY RESULTS

In a study like ours, the emphasis is not on measuring raw performance, but rather on characterising the diversity of behaviours arising from the spectrum of proposed variants. A starting point is to return to Figure 1 (right), and assess how much of the previously untouched space is now filled

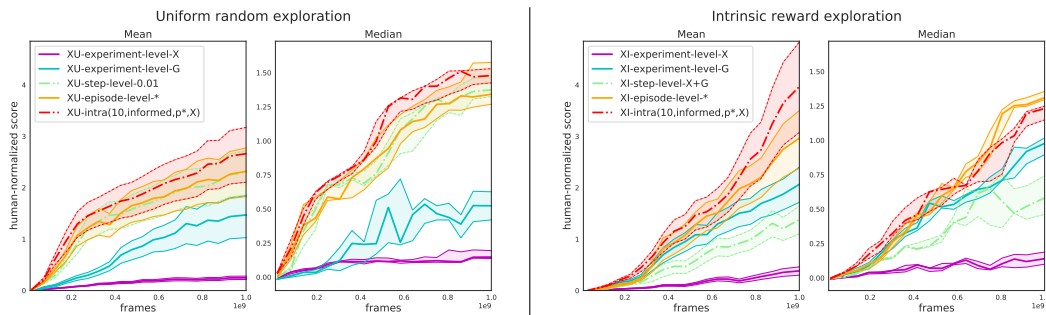

Figure 3: Human-normalized performance results aggregated over 7 Atari games and 3 seeds, comparing the four levels of exploration granularity. **Left two**: uniform explore mode $\mathcal{X}_U$. **Right two**: RND intrinsic reward explore mode $\mathcal{X}_I$. In each case, the baselines are pure modes $\mathcal{X}$ and $\mathcal{G}$, step-level switching with $\varepsilon$-greedy, and episodic switching (with a bandit-adapted proportion). Note that the $\mathcal{X}_I$-step-level experiment uses both an intrinsic and an extrinsic reward, as in Burda et al. (2018).

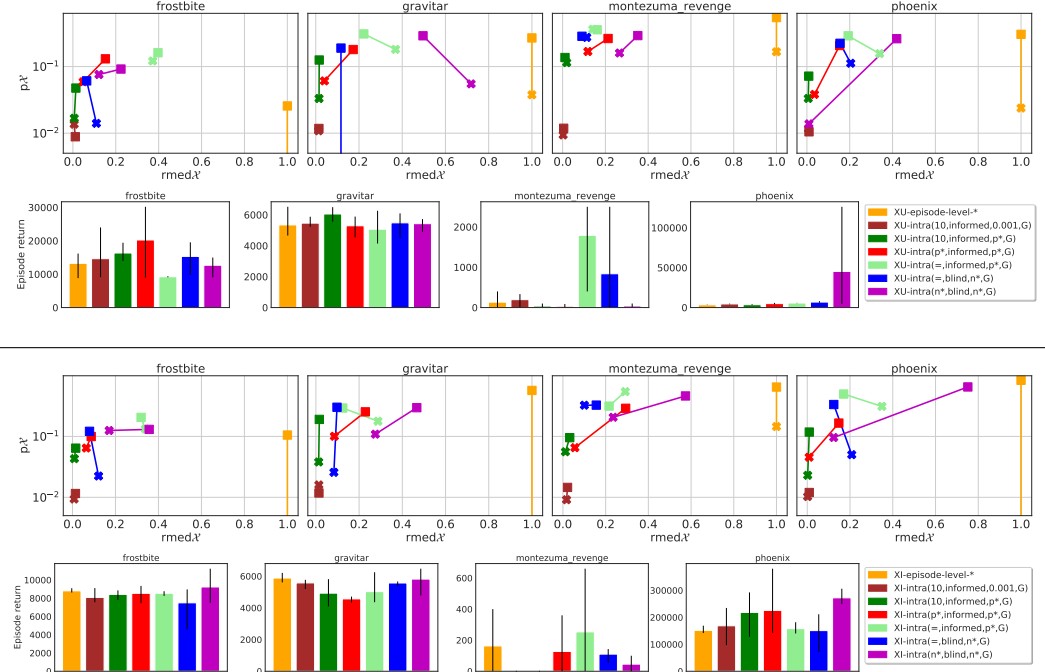

Figure 4: **Rows 1 and 3**: Summary characteristics $p_\mathcal{X}$ and $\mathrm{rmed}_\mathcal{X}$ of induced exploration behaviour, for different variants of intra-episodic exploration (and an episodic baseline for comparison), on a subset of 4 Atari games. Bandit adaptation can change these statistics over time, hence square and cross markers show averages over first and last $10\%$ of training, respectively. **Rows 2 and 4**: Corresponding final scores (averaged over final $10\%$ of training). Error bars show the span between $\min$ and $\max$ performance across 3 seeds. Note how different variants cover different parts of characteristic space, and how the bandit adaptation shifts the statistics into different directions for different games. See main text for further discussion and Appendix C for other games and variants.

by intra-episodic variants, and how the 'when' characteristics translate into performance. Figure 4 answers these questions, and raises some new ones. First off, the raw amount of exploration $p_\mathcal{X}$ is not a sufficient predictor of performance, implying that the temporal structure matters. It also shows substantial bandit adaptation at work: compare the exploration statistics at the start (squares) and end-points of training (crosses), and how these trajectories differ per game; a common pattern is that reducing $p_\mathcal{X}$ far below $0.5$ is needed for high performance. Interestingly, these adaptations are similar between $\mathcal{X}_U$ and $\mathcal{X}_I$, despite very different explore modes (and differing performance

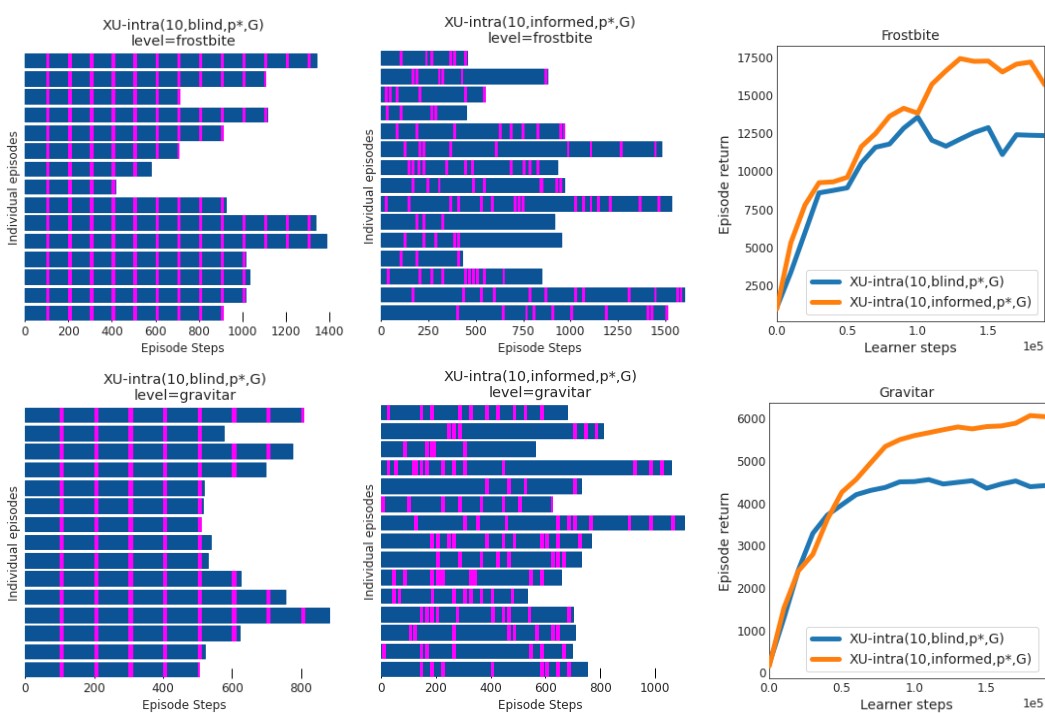

Figure 5: **Left and center**: Illustration of detailed temporal structure within individual episodes, on FROSTBITE (top) and GRAVITAR (bottom), contrasting two trigger mechanisms. Each subplot shows 15 randomly selected episodes (one per row) that share the same overall exploration amount $p_\mathcal{X} = 0.1$. Each vertical bar (magenta) represents an exploration period of fixed length $n_\mathcal{X} = 10$; each blue chunk represents an exploitation period. **Left**: blind, step-based trigger leads to equally spaced exploration periods. **Center**: a trigger signal informed by value promise leads to very different within-episode patterns, with some parts being densely explored, and others remaining in exploit mode for very long. **Right**: the corresponding learning curves show a clear performance benefit for the informed trigger variant (orange) in this particular setting. Appendix C has similar plots for many more variants and games.

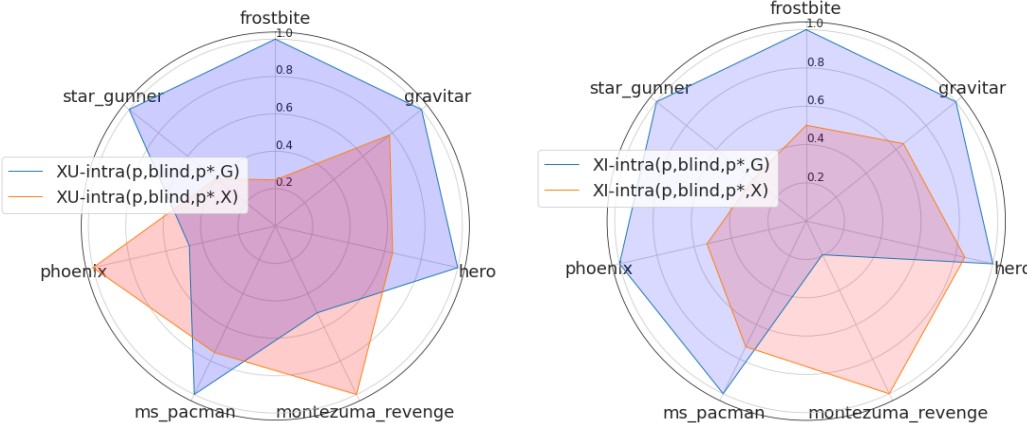

Figure 6: Starting mode effect. Final mean episode return for two blind intra-episode experiments that differ only in start mode, greedy (blue) or explore (orange). Scores are normalised so that 1 is the maximum result across the two start modes. Either choice can reliably boost or harm performance, depending on the game. **Left**: uniform explore mode $\mathcal{X}_U$. **Right**: intrinsic reward explore mode $\mathcal{X}_I$.

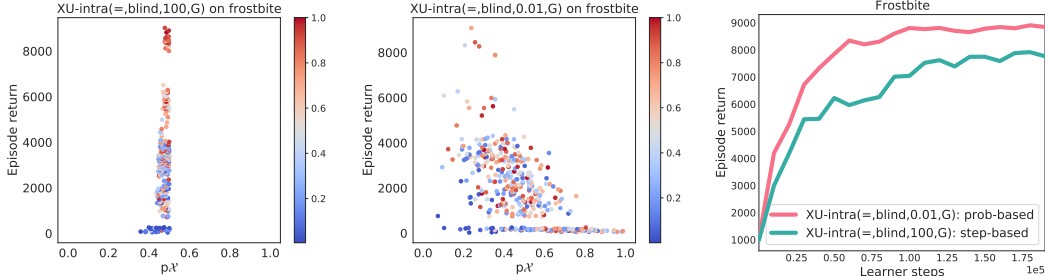

Figure 7: **Left and center**: Contrasting the behavioural characteristics between two forms of blind switching, step-based (left) and probabilistic (center), on the example of FROSTBITE. Each point is an actor episode, with colour indicating time in training (blue for early, red for late). Note the higher diversity of $p_\mathcal{X}$ when switching probabilistically. **Right**: Corresponding performance curves indicate that the probabilistic switching (red) has a performance benefit, possibly because it creates the opportunity for 'lucky' episodes with much less randomness in a game where random actions can easily kill the agent. For more games, please see the Appendix C.

results). We would expect prolonged intrinsic exploration periods to be more useful than prolonged random ones, and indeed, comparing the high-$\mathrm{rmed}_\mathcal{X}$ variant (purple) across $\mathcal{X}_U$ and $\mathcal{X}_I$, it appears more beneficial for the latter. Zooming in on specific games, a few results stand out: in $\mathcal{X}_U$ mode, the only variant that escapes the inherent local optimum of PHOENIX is the blind, doubly adaptive one (purple), with the bandits radically shifting the exploration statistics over the course of training. In contrast, the best results on MONTEZUMA'S REVENGE are produced by the symmetric, informed trigger variant (light green), which is forced to retain a high $p_\mathcal{X}$. Finally, FROSTBITE is the one game where an informed trigger (red) clearly outperforms its blind equivalent (purple).

These insights are still limited to summary statistics, so Figure 5 looks in more depth at the detailed temporal structure within episodes (as in Figure 1, left). Here the main comparison is between blind and informed triggers, illustrating that the characteristics of the fine-grained within-episode structure can differ massively, despite attaining the same high-level statistics $p_\mathcal{X}$ and $\mathrm{med}_\mathcal{X}$. We can see quite a lot of variation in the trigger structure – the moments we enter exploration are not evenly spaced anymore. As a bonus, the less rigid structure of the informed trigger (and possibly the more carefully chosen switch points) end up producing better performance too.

Figure 6 sheds light on a complementary dimension, differentiating the effects of starting in explore or exploit mode. In brief, each of these can be consistently beneficial in some games, and consistently harmful in others. Another observation here is the dynamics of the bandit adaptation: when starting in exploit mode, it exhibits a preference for long initial exploit periods in many games (up to 10000 steps), but that effect vanishes when starting in explore mode (see also Appendix C). More subtle effects arise from the choice of parameterisation of switching rates. Figure 7 shows a stark qualitative difference on how probabilistic switching differs from step-count based switching, with the former spanning a much wider diversity of outcomes, which improves performance.

### 3.4 TAKE-AWAYS

Summarising the empirical results in this section, two messages stand out. First, there seems to be a sweet spot in terms of temporal granularity, and intra-episodic exploration is the right step towards finding it. Second, the vastly increased design space of our proposed family of methods gives rise to a large diversity of behavioural characteristics; and this diversity is not superficial, it also translates to meaningful performance differences, with different effects in different games, which cannot be reduced to simplistic metrics, such as $p_\mathcal{X}$. In addition, we provide some sensible rules-of-thumb for practitioners willing to join us on the journey of intra-episodic exploration. In general, it is useful to let a bandit figure out the precise settings, but it is worth curating its choices to at most a handful. Jointly using two bandits across factored dimensions is very adaptive, but can sometimes be harmful when they decrease the signal-to-noise ratio in each other's learning signal. Finally, the choice of the uncertainty-based trigger should be informed by the switching modes (see Appendix B for details).

## 4 DISCUSSION

**Time-based exploration control** The emphasis of our paper was on the benefits of heterogeneous temporal structure in mode-switching exploration. Another potential advantage over monolithic approaches is that it may be easier to tune hyper-parameters related to an explicit exploration budget (e.g., via $p_\mathcal{X}$) than to tune an intrinsic reward coefficient, especially if extrinsic reward scales change across tasks or time, and if the non-stationarity of the intrinsic reward affects its overall scale.

**Diversity for diversity's sake** One role of a general-purpose exploration method is to allow an agent to get off the ground in a wide variety of domains. While this may clash with sample-efficient learning on specific domains, we believe that the former objective will come to dominate in the long run. In this light, methods that exhibit more diverse behaviour are preferable for that reason alone because they are more likely to escape local optima or misaligned priors.

**Related work** While not the most common approach to exploration in RL, we are aware of some notable work investigating non-trivial temporal structure. The $\epsilon z$-greedy algorithm (Dabney et al., 2020) initiates contiguous chunks of directed behaviour ('flights') with the length sampled from a heavy-tailed distribution. In contrast to our proposal, these flights act with a single constant action instead of invoking an explore mode. Campos et al. (2021) pursue a similar idea, but with flights along pre-trained coverage policies, while Ecoffet et al. (2021) chain a 'return-to-state' policy to an explore mode. Maybe closest to our $\mathcal{X}_I$ setting is the work of Bagot et al. (2020), where periods of intrinsic reward pursuit are explicitly invoked by the agent. Exploration with *gradual* change instead of abrupt mode switches generally appears at long time-scales, such as when pursuing intrinsic rewards (Schmidhuber, 2010; Oudeyer & Kaplan, 2009) but can also be effective at shorter time-scales, e.g., Never-Give-Up (Badia et al., 2020b). Related work on the question of which states to prefer for exploration decisions (Tokic, 2010) tends not to consider prolonged exploratory periods.

**Relation to options** Ideas related to switching behaviours at intra-episodic time scales are well-known outside of the context of exploration. In the *options* framework in hierarchical RL, the goal is to chain together a sequence of sub-behaviours into a reward-maximising policy (Sutton et al., 1999; Mankowitz et al., 2016). Some work has looked at using options for exploration too (Jinnai et al., 2019a; Bougie & Ichise, 2021). In its full generality, the options framework is a substantially more ambitious endeavour than our proposal, as it requires learning a full state-dependent hierarchical policy that picks which option to start (and when), as well as jointly learning the options themselves.

**Limitations** Our proposed approach inherits many typical challenges for exploration methods, such as sample efficiency or trading off risk. An aspect that is particular to the intra-episode switching case is the different nature of the off-policy-ness. The resulting effective policy can produce state distributions that differ substantially from those of either of the two base mode behaviours that are being interleaved. It can potentially visit parts of the state space that neither base policy would reach if followed from the beginning of the episode. While a boon for exploration, this might pose a challenge to learning, as it could require off-policy corrections that treat those states differently and do not only correct for differences in action space. Our paper does not use (non-trivial) off-policy correction (see Appendix A) as our initial experimentation showed it is not an essential component in the current setting (see Figure 15). We leave this intriguing finding for future investigation.

**Future work** With the dimensions laid out in Section 2, it should be clear that this paper can but scratch the surface. We see numerous opportunities for future work, some of which we already carried out initial investigations (see Appendix B). For starters, the mechanism could go beyond two-mode and switch between exploit, explore, novelty and mastery (Thomaz & Breazeal, 2008), or between many diverse forms of exploration, such as levels of optimism (Derman et al., 2020; Moskovitz et al., 2021). Triggers could be expanded or refined by using different estimations of uncertainty, such as ensemble discrepancy (Wiering & Van Hasselt, 2008; Buckman et al., 2018), amortised value errors (Flennerhag et al., 2020), or density models (Bellemare et al., 2016; Ostrovski et al., 2017); or other signals, such as salience (Downar et al., 2002), minimal coverage (Jinnai et al., 2019a;b), or empowerment (Klyubin et al., 2005; Gregor et al., 2016; Houthooft et al., 2016).

**Conclusion** We have presented an initial study of intra-episodic exploration, centred on the scenario of switching between an explore and an exploit mode. We hope this has broadened the available forms of temporal structure in behaviour, leading to more diverse, adaptive and intentional forms of exploration, in turn enabling RL to scale to ever more complex domains.

## ACKNOWLEDGMENTS

We would like to thank Audrunas Gruslys, Simon Osindero, Eszter Vértes, David Silver, Dan Horgan, Zita Marinho, Katrina McKinney, Claudia Clopath, David Amos, Víctor Campos, Remi Munos, and the entire DeepMind team for discussions and support, and especially Pablo Sprechmann and Luisa Zintgraf for detailed feedback on an earlier version.

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

# A    DETAILED EXPERIMENTAL SETUP

## A.1    ATARI ENVIRONMENT

We use a selection of games from the widely used Atari Learning Environment (ALE, Bellemare et al. (2013)). It is configured to not expose the 'life-loss' signal, and use the full action set (18 discrete actions) for all games (not the per-game reduced effective action spaces). We also use the *sticky*-action randomisation as in (Machado et al., 2018). Episodes time-out after 108k frames (i.e. 30 minutes of real-time game play).

Differently from most past Atari RL agents following DQN (Mnih et al., 2015), our agent uses the raw $210 \times 160$ RGB frames as input to its value function (one at a time, without frame stacking), though it still applies a max-pool operation over the most recent 2 frames to mitigate flickering inherent to the Atari simulator. As in most past work, an action-repeat of 4 is applied, over which rewards are summed.

## A.2    AGENT

The agent used in our Atari experiments is a distributed implementation of a value- and replay-based RL algorithm derived from the Recurrent Replay Distributed DQN (R2D2) architecture (Kapturowski et al., 2019). This system comprises of a fleet of 120 CPU-based actors (combined with a single TPU for batch inference) concurrently generating experience and feeding it to a distributed experience replay buffer, and a single TPU-based learner randomly sampling batches of experience sequences from replay and performing updates of the recurrent value function by gradient descent on a suitable RL loss.

The value function is represented by a convolutional torso feeding into a linear layer, followed by a recurrent LSTM (Hochreiter & Schmidhuber, 1997) core, whose output is processed by a further linear layer before finally being output via a Dueling value head (Wang et al., 2016). The exact parameterisation follows the slightly modified R2D2 presented by Dabney et al. (2020) and Schaul et al. (2021). Refer to Table 1 for a full list of hyper-parameters. It is trained via stochastic gradient descent on a multi-step TD loss (more precisely, a 5-step Q-learning loss) with the use of a periodically updated target network (Mnih et al., 2015) for bootstrap target computation, using mini-batches of sampled replay sequences. Replay sampling is performed using prioritized experience replay (Schaul et al., 2016) with priorities computed from sequences' TD errors following the scheme introduced by Kapturowski et al. (2019). As in R2D2, sequences of 80 observations are used for replay, with a prefix of 20 observations used for burn-in. In a slight deviation from the original, our agent uses a fixed replay ratio of 1, i.e. the learner or actors get throttled dynamically if the average number of times a sample gets replayed exceeds or falls below this value; this makes experiments more reproducible and stable.

Actors periodically pull the most recent network parameters from the learner to be used in their exploratory policy. In addition to feeding the replay buffer, all actors periodically report their reward, discount and return histories to the learner, which then calculates running estimates of reward, discount and return statistics to perform return-based scaling (Schaul et al., 2021). If applicable, the episodic returns from the actors are also sent to the non-stationary bandit(s) that adapt the distribution over exploration parameters (e.g., target ratios $\rho$ or period lengths $n_\mathcal{X}$). In return, the bandit(s) provide samples from that distribution to each actor at the start of a new episode, just like Schaul et al. (2019).

Our agent is implemented with JAX (Bradbury et al., 2018), uses the Haiku (Hennigan et al., 2020), Optax (Budden et al., 2020b), Chex (Budden et al., 2020a), and RLax (Hessel et al., 2020) libraries for neural networks, optimisation, testing, and RL losses, respectively, and Reverb (Cassirer et al., 2020) for distributed experience replay.

## A.3    TRAINING AND EVALUATION PROTOCOLS

All our experiments ran for 200k learner updates. With a replay ratio of 1, sequence length of 80 (adjacent sequences overlapping by 40 observations), a batch size of 64, and an action-repeat of 4 this corresponds to a training budget of $200000 \times 64 \times 40 \times 1 \times 4 \approx 2\text{B}$ environment frames (which

| **Neural Network** | |
|---|---|
| Convolutional torso channels | $32, 64, 128, 128$ |
| Convolutional torso kernel sizes | $7, 5, 5, 3$ |
| Convolutional torso strides | $4, 2, 2, 1$ |
| Pre-LSTM linear layer units | $512$ |
| LSTM hidden units | $512$ |
| Post-LSTM linear layer units | $256$ |
| Dueling value head units | $2 \times 256$ (separate linear layer for each of value and advantage) |
| **Acting** | |
| Initial random No-Ops | None |
| Sticky actions | Yes (prob 0.25) |
| Action repeats | 4 |
| Number of actors | 120 |
| Actor parameter update interval | 400 environment steps |
| **Replay** | |
| Replay sequence length | 80 (+ prefix of 20 of burn-in) |
| Replay buffer size | $4 \times 10^6$ observations ($10^5$ part-overlapping sequences) |
| Priority exponent | 0.9 |
| Importance sampling exponent | 0.6 |
| Fixed replay ratio | 1 update per sample (on average) |
| **Learning** | |
| Multi-step Q-learning | $k = 5$ |
| Off-policy corrections | None |
| Discount $\gamma$ | 0.997 |
| Reward clipping | None |
| Return-based scaling | as in (Schaul et al., 2021) |
| Mini-batch size | 64 |
| Optimizer & settings | Adam (Kingma & Ba, 2014), learning rate $\eta = 2 \times 10^{-4}$, $\epsilon = 10^{-8}$, momentum $\beta_1 = 0.9$, second moment $\beta_2 = 0.999$ |
| Gradient norm clipping | 40 |
| Target network update interval | 400 updates |
| **RND settings** | |
| Convolutional torso channels | $32, 64, 64$ |
| Convolutional torso kernel sizes | $8, 4, 3$ |
| Convolutional torso strides | $4, 2, 1$ |
| MLP hidden units | 128 |
| Image downsampling stride | $2 \times 2$ |

Table 1: Hyper-parameters and settings.

is less than 10% of the original R2D2 budget). In wall-clock-time, one such experiment takes about 12 hours using 2 TPUs (one for the batch inference, the other for the learner) and 120 CPUs.

For evaluation, a separate actor (not feeding the replay buffer) is running alongside the agent using a greedy policy ($\varepsilon = 0$), and pulling the most recent parameters at the beginning of each episode. We follow standard evaluation methodology for Atari, reporting mean and median 'human-normalised' scores as introduced in (Mnih et al., 2015) (i.e. the episode returns are normalised so that 0 corresponds to the score of a uniformly random policy while 1 corresponds to human performance), as well as the mean 'human-capped' score which caps the per-game performance at human level. Error bars or shaded curves correspond to the minimum and maximum values across these seeds.

## A.4    RANDOM NETWORK DISTILLATION

The agent setup for the $\mathcal{X}_I$ experiments differs in a few ways from the default described above. First, a separate network is trained via Random Network Distillation (RND, (Burda et al., 2018)), which consists of a simple convnet with an MLP (no recurrence); for detailed settings, see RND section in Table 1. The RND prediction network is updated jointly with the Q-value network, on the same

data. The intrinsic reward derived from the RND loss is pursued at the same discount $\gamma = 0.997$ as the external reward in $\mathcal{G}$. The Q-value network is augmented with a *second head* that predicts the Q-values for the intrinsic reward; this branches off after the 'Post-LSTM linear layer' (with 256), and is the same type of dueling head, using the same scale normalisation method (Schaul et al., 2021). In addition, the 5-step Q-learning is adapted to use a simple off-policy correction, namely trace-cutting on non-greedy actions (akin to Watkins Q($\lambda$) with $\lambda = 1$), separately for each learning head.[3] The $\mathcal{X}_I$ policy is the greedy policy according to the Q-values of the second head. Note that because of these differences in set-up, and especially because the second head can function as an auxiliary learning target, it may be misleading to compare $\mathcal{X}_I$ and $\mathcal{X}_U$ results head-to-head: we recommend looking at how things change within one of these settings (across variants of intra-episodic exploration or the baselines), rather than between them.

## A.5 HOMEOSTASIS

The role of the homeostasis mechanism is to transform a sequence of scalar signals $x_t \in \mathbb{R}$ (for $1 \leq t \leq T$) into a sequence of binary switching decisions $y_t \in \{0, 1\}$ so that the average number of switches approximates a desired target rate $\rho$, that is , $\frac{1}{T} \sum_t y_t \approx \rho$, and high values of $x_t$ correspond to a higher probability of $y_t = 1$. Furthermore, the decision at any point $y_t$ can only be based on the past signals $x_{1:t}$. One way to achieve this is to exponentiate $x$ (to turn it into a positive number $x^+$) and then set an adaptive threshold to determine when to switch. Algorithm 1 describes how this is done in pseudo-code. The implementation defines a time-scale of interest $\tau := \min(t, 100/\rho)$, and uses it to track moving averages of three quantities, namely the mean and variance of $x$, as well as the mean of $x^+$.

---

**Algorithm 1** Homeostasis

---

**Require:** target rate $\rho$
 1: initialize $\overline{x} \leftarrow 0, \overline{x^2} \leftarrow 1, \overline{x^+} \leftarrow 1$
 2: **for** $t \in \{1, \ldots, T\}$ **do**
 3:     obtain next scalar signal return $x_t$
 4:     set time-scale $\tau \leftarrow \min(t, \frac{100}{\rho})$
 5:     update moving average $\overline{x} \leftarrow (1 - \frac{1}{\tau})\overline{x} + \frac{1}{\tau}x_t$
 6:     update moving variance $\overline{x^2} \leftarrow (1 - \frac{1}{\tau})\overline{x^2} + \frac{1}{\tau}(x_t - \overline{x})^2$
 7:     standardise and exponentiate $x^+ \leftarrow \exp\left(\frac{x_t - \overline{x}}{\sqrt{\overline{x^2}}}\right)$
 8:     update transformed moving average $\overline{x^+} \leftarrow (1 - \frac{1}{\tau})\overline{x^+} + \frac{1}{\tau}x^+$
 9:     sample $y_t \sim \text{Bernoulli}\left(\min\left(1, \rho\frac{x^+}{\overline{x^+}}\right)\right)$
10: **end for**

---

In our informed trigger experiments we use value promise as the particular choice of trigger signal $x_t = D_{\text{promise}}(t - k, t)$. As discussed in Section 3.1, when using a bandit, its choices for target rates are $\rho \in \{0.1, 0.01, 0.001, 0.0001\}$.

## B OTHER VARIANTS

The results we report in the main paper are but a subset of the possible variants that could be tried in this rather large design space. In fact, we have done initial investigations on a few of these, which we report below.

### B.1 ADDITIONAL EXPLORE MODES

**Softer explore-exploit modes** The all-or-nothing setting with a greedy exploit mode and a uniform random explore mode is clear and simple, but it is plausible that less extreme choices could work well too, such as an $\varepsilon$-greedy explore mode with $\varepsilon = 0.4$ and an $\varepsilon$-greedy exploit mode with

---

[3]While this seemed like important aspect to us, it turned out to make very little difference in hindsight, see Figure 15.

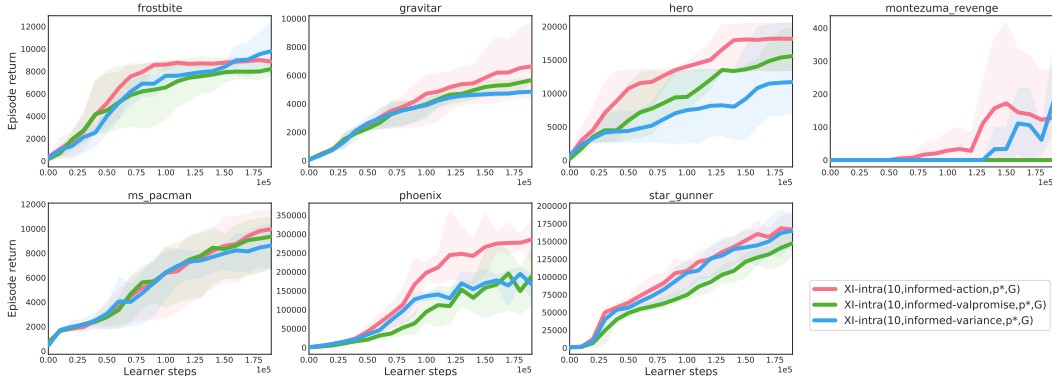

Figure 8: Preliminary results comparing different informed triggers: value-discrepancy, action-mismatch, and variance-based, when using $\mathcal{X}_I$ exploration mode.

$\varepsilon = 0.1$. We denote this pairing as $\mathcal{X}_S$. Preliminary results (see Figure 16) indicate that overall performance is mostly similar to $\mathcal{X}_U$, possibly less affected by the choice of granularity and triggers.

**Different discounts**  Another category of explore mode ($\mathcal{X}_\gamma$) is to pursue external reward but at a different time-scale (e.g., a much shorter discount like $\gamma = 0.97$). This results in less of a switch between explore and exploit modes, but rather in an alternation of long-term and short-term reward pursuits, producing a different kind of behavioural diversity. So far, we do not have conclusive results to report with this mode.

### B.2 ADDITIONAL INFORMED TRIGGERS

**Action-mismatch-based triggers**  Another type of informed trigger is to derive an uncertainty estimate from the discrepancies across an ensemble. For example, we can train two heads that use an identical Q-learning update but are initialised differently. From that, we can measure multiple forms of discrepancy, a nice and robust one is to rank the actions according to each head and compute how large the overlap among the top-$k$ actions is.

**Variance-based triggers**  Another type of informed trigger is to measure the variance of the Q-values themselves, taken across such an ensemble (of two heads) and use that as an alternative uncertainty-based trigger.

Figure 8 shows preliminary results on how performance compares across these two new informed triggers, in relation to the value-promise one from Section 2.2. Overall, the action-mismatch trigger seems to have an edge, at least in this setting, and we plan to investigate this further in the future. From other probing experiments, it appears that for other explore modes, different trigger signals are more suitable.

## C ADDITIONAL RESULTS

This section includes additional results. Wherever the main figures included a subset of games or variants (Figures 4, 5, 7) we show full results here (Figures 11, 12, 13, respectively), and the aggregated performances of Figure 3 are split out into individual games in Figure 10. Also, some of the learning curves from Figures 4 and 11 are shown in Figure 16. In addition, Figure 14 illustrates how the internal bandit probabilities evolve over time based on starting mode for the experiments shown in Figure 6. Lastly, we provide some bonus illustrations for the intra-episodic design decisions included in the namings of our variants in Figure 9 with the hope of facilitating the interpretation and intuition for intra-episodic variants and their resulting behaviours.

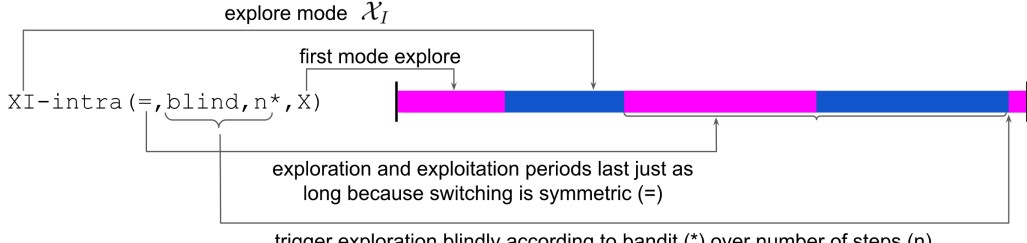

Figure 9: Extra example illustrating the space of design decisions for intra-episodic exploration.

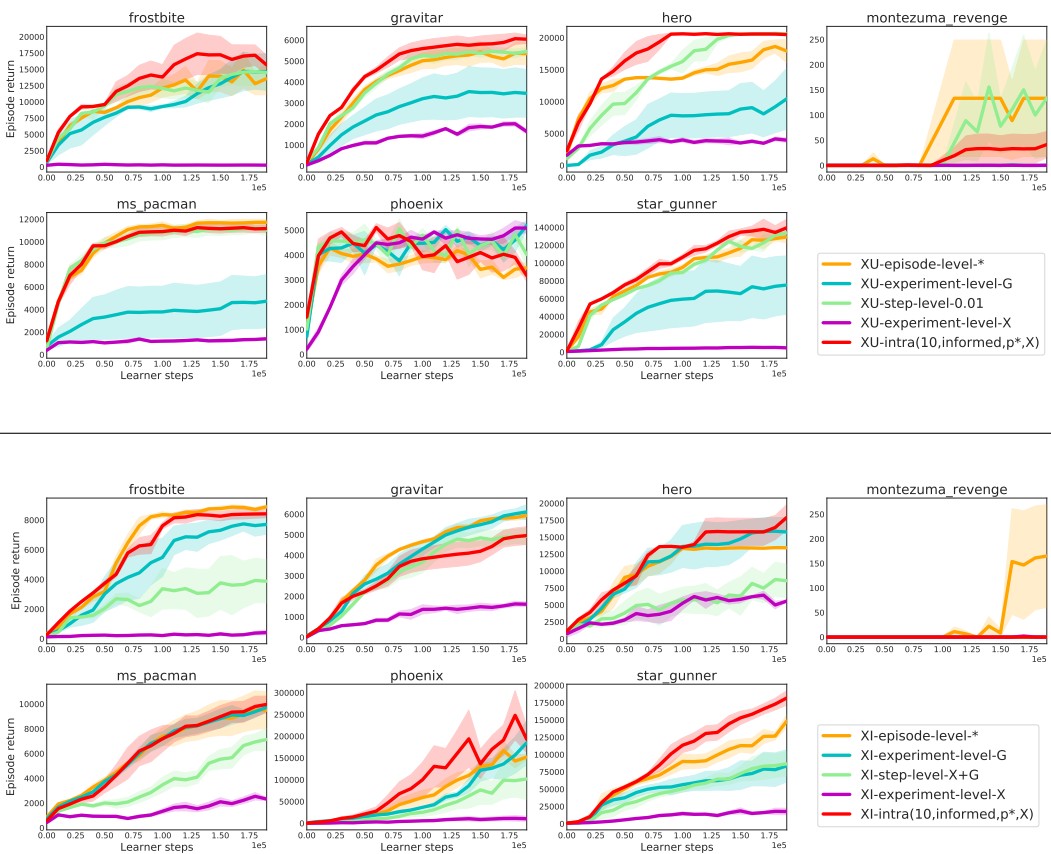

Figure 10: Extension of Figure 3, showing performance results as the mean episode return (with error bars spanning the min and max performance across 3 seeds) for the 7 Atari games tried. We compare the four levels of exploration granularity as described in section 2.1 for $\mathcal{X}_U$ mode (top two rows) and $\mathcal{X}_I$ mode (bottom two rows). Intra-episodic switching (red curve) is superior or comparable to existing, monolithic or well-established approaches. Note that here we compare with the same intra-episodic switching mechanism (i.e., with `XU-` or `XI-intra(10,informed,p*,X)`), but there is at least one intra-episodic variant for each game which results in clear performance gains (just that it is not the same intra-episodic variant across games or as the one shown here).

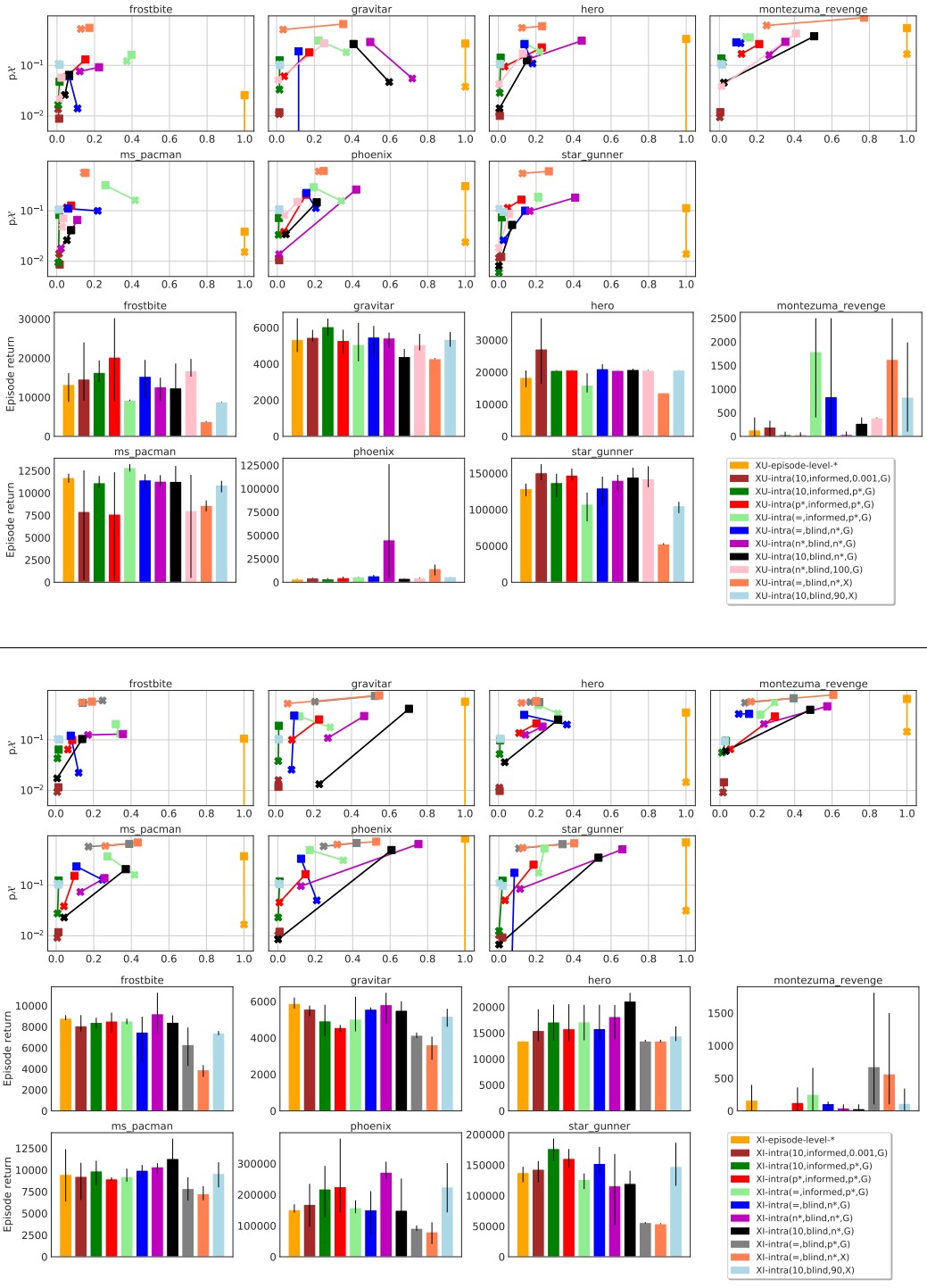

Figure 11: Extension of figure 4 to all Atari games and intra-episodic switching variants tried. We show the characteristic space of exploration (summarized by $\mathrm{rmed}_{\mathcal{X}}$ and $p_{\mathcal{X}}$ on the X and Y axis, respectively) on rows 1, 2, 5, and 6, and how different explore-exploit proportions translate to performance (error bars spanning the $\min$ and $\max$ performance across 3 seeds) on rows 3, 4, 7, and 8, for $\mathcal{X}_U$ mode (top) and $\mathcal{X}_I$ mode (bottom). Note how different intra-episodic switching variants cover different parts of characteristic space and how the meta-controller adapts and changes the exploration statistics over time. This figure shows how fine-grained and varied intra-episodic switching can be, and how it translates to rich, diverse, and beneficial behaviours.

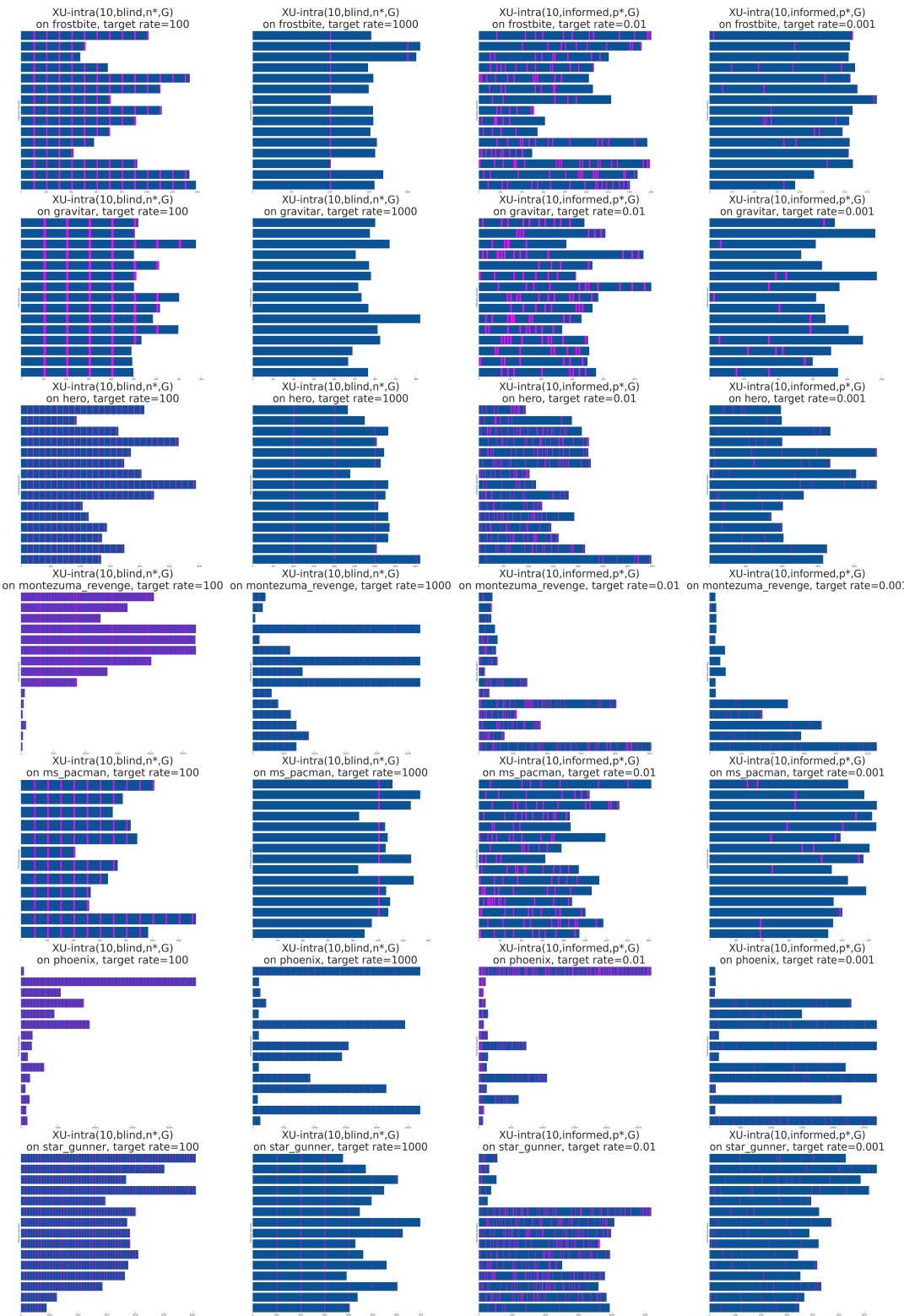

Figure 12: Extension of Figure 5 to the 7 Atari games we experimented with. **First two columns:** temporal structures for a blind, step-based trigger; the 15 episodes we randomly selected correspond to 100 and 1000 fixed switching steps; the exploration period was fixed to 10 steps. **Last two columns:** temporal structures obtained with an equivalent informed trigger and corresponding to target rates of 0.01 and 0.001, respectively.

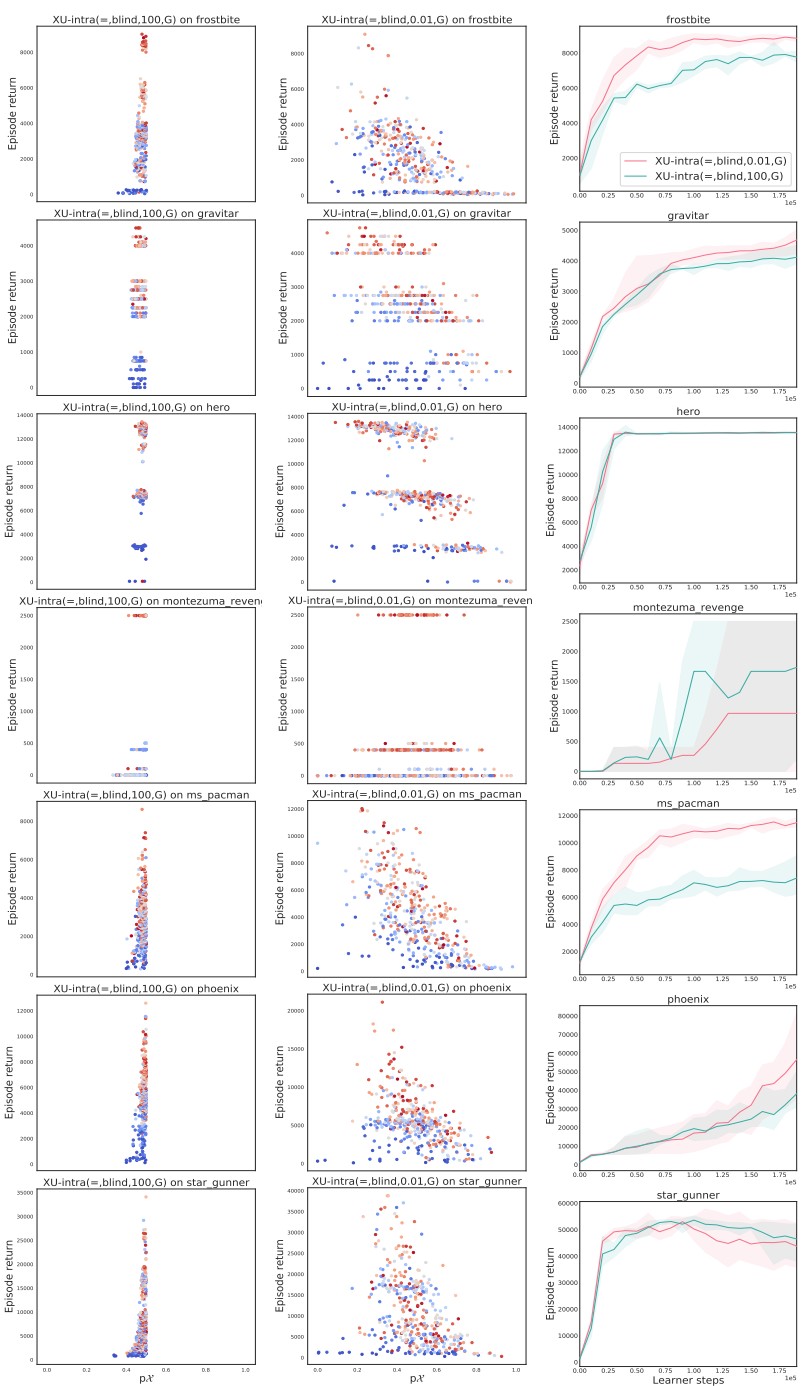

Figure 13: Extension of Figure 7, showing behavioural characteristics (exploration proportion $p_\mathcal{X}$) between two forms of blind switching, step-based (left) and probabilistic (center), with their corresponding performances (right).

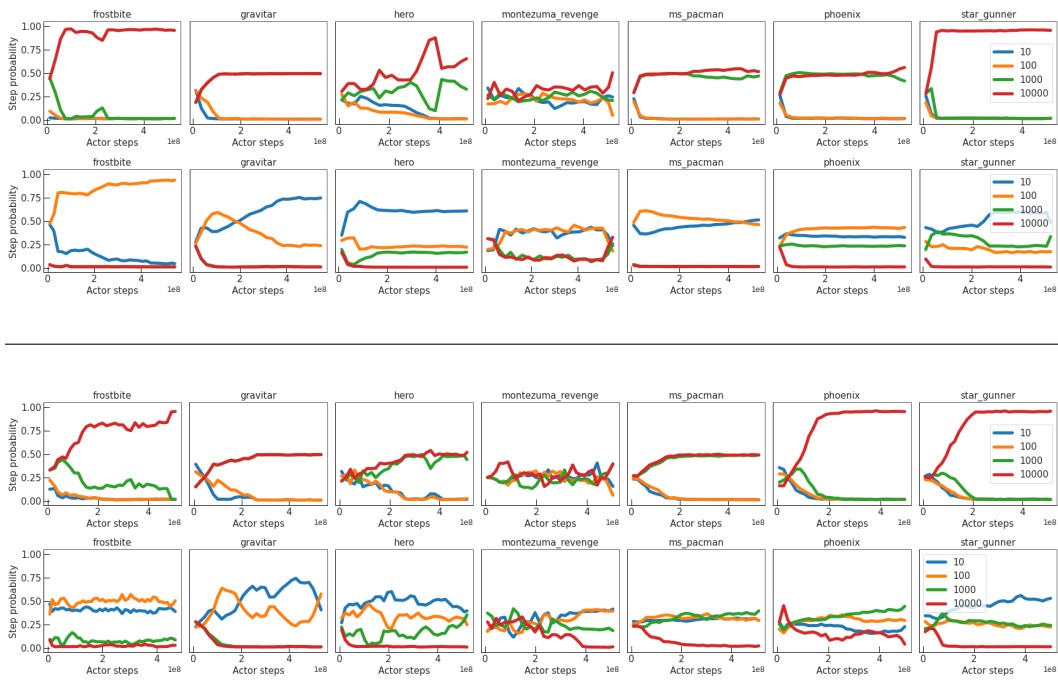

Figure 14: Extension of Figure 6, showing the performance differences between two blind intra-episode experiments, starting either in explore ($\mathcal{X}$, rows 2 and 4) or in exploit mode ($\mathcal{G}$, rows 1 and 3). We show the bandit arm probabilities for each of the step sizes $n_{\mathcal{X}}$ and how they change over the course of learning for $\mathcal{X}_U$ (top two rows) and for $\mathcal{X}_I$ modes (bottom two rows). **Findings**: for symmetric blind triggers, starting with exploitation results in slower rates of switching (high $n_{\mathcal{X}} = n_{\mathcal{G}}$ like red and green); in contrast, starting with exploration results in behaviours promoting higher switching rates (small $n_{\mathcal{X}} = n_{\mathcal{G}}$ like blue and orange). Note that these preferences are not matching perfectly across all games, and thus results are domain-dependent.

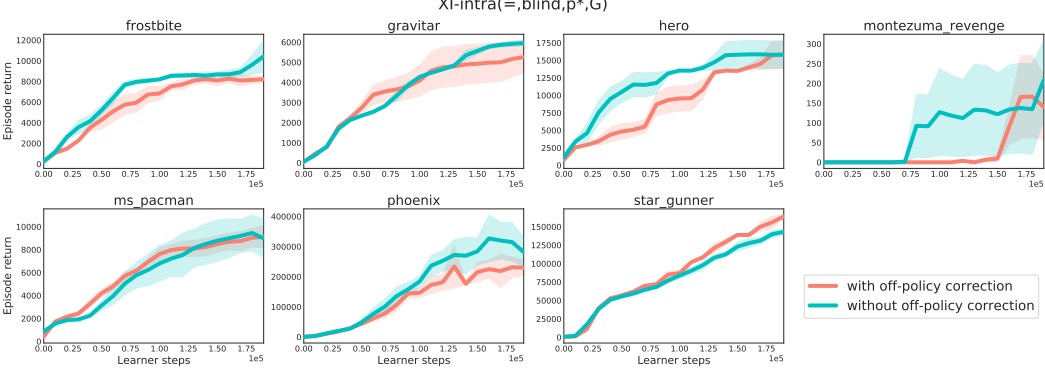

Figure 15: Results of a probe experiment around off-policy correction in $\mathcal{X}_I$ mode. Red is Watkins $Q(\lambda)$ with $\lambda = 1$ while blue is uncorrected $k$-step $Q$-learning, in each case with returns of length at most $k = 5$. The performance is the same (or even slightly better) **without** off-policy correction, showing that it is not critical in our current setting.

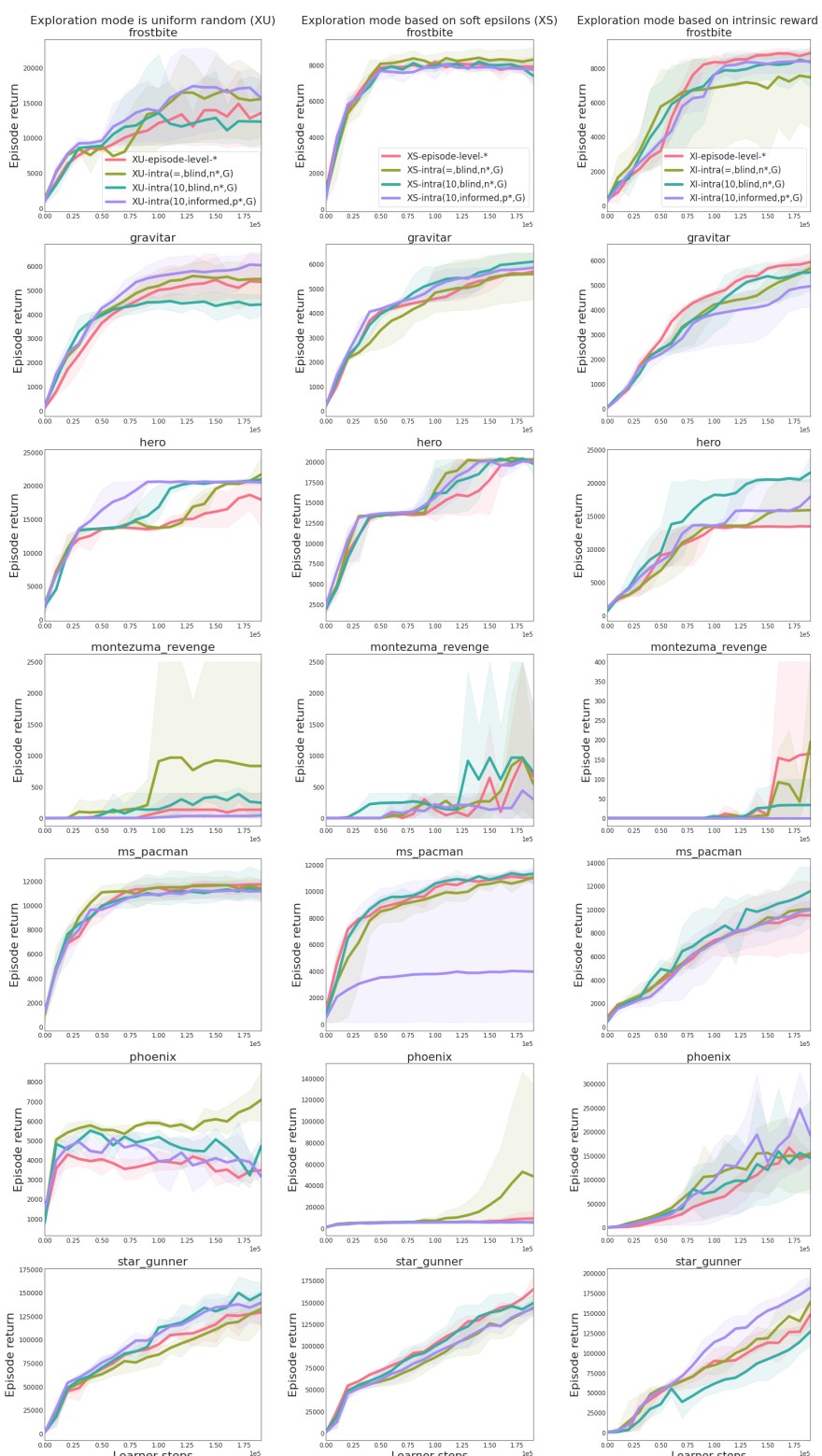

Figure 16: Comparing 3 different $\mathcal{X}$ modes on the same 4 experimental settings and across 7 Atari games: uniform exploration ($\mathcal{X}_U$, left), soft-epsilon-based exploration ($\mathcal{X}_S$, center), and intrinsic exploration ($\mathcal{X}_I$, right).

