# OpenReview forum: "When should agents explore?"
_ICLR.cc/2022/Conference — ICLR 2022 Spotlight_

### Official Review · Reviewer_AkpW · 2021-11-01

**Correctness:** 3
**Technical Novelty And Significance:** 4
**Empirical Novelty And Significance:** 4
**Recommendation:** 6
**Confidence:** 3

**Main Review:**

The idea of mode-switching is interesting. However, the presentation of the motivation as well as the results seem somewhat weak in my view. The main comparison baseline for the mode-switching architecture is the monolithic variant, where typically sparse rewards in hard exploration tasks are augmented with intrinsic reward signals. This means that the modes that they switch between in this paper are merged homogeneously in time. A valid problem that the authors point out for the monolithic case is that the scale of the intrinsic reward signal has to be tuned and may need to change in time. But there is no comparison to these methods and the superiority of their method to the monolithic variants are not highlighted well enough.

The authors try to circumvent the performance comparisons with other baselines by saying that they can obtain more diverse behaviours (in terms of exploration strategies?) and say that they don’t aim to show improved performance. However, this diversity argumentation is not strong enough in my view. First, it is not clear to me if the authors want to focus on the diversity of behaviour that is obtained within one variant of the mode-switching, e.g. informed switching, or whether they want to highlight the diversity across different granularities and switching mechanisms. If it is the latter, the diverse behaviour doesn’t necessarily translate into performance for the different games. And best performance for different games might be obtained with different strategies, but since this strategy is fixed prior to the experiment, it is not clear to me how this helps the diversity argument of the method itself. The authors point out that Montezuma’s Revenge and Phoenix have their best performance with different mode-switching behaviours, but these results have a really high variance. Perhaps, 3 seeds are just not enough and more seeds are needed to draw reliable conclusions.

**Summary Of The Paper:**

This paper proposes a mode-switching strategy for the exploration/exploitation dilemma instead of monolithic behaviour policies in order to obtain more diverse behaviour. Different granularities for the timing of the switches as well as different switching mechanisms are investigated (blind vs. informed switching). The focus for exploration is not on how, but when. For exploration, they both use Random Network Distillation (RND) as well as a uniform policy. Their experiments are conducted on the Atari Learning Environment (ALE), where they provide performance and diversity results.

**Summary Of The Review:**

I believe that the idea presented in this paper is interesting but the results are lacking. The related work section briefly covers some similar methods, and I think comparisons are still needed with these other methods. E.g. GoExplore also focuses on the *when* question of exploration and should be included as a baseline as well as works with monolithic behaviour policies where the mode-switching is replaced by a weighting problem of external and intrinsic rewards of a single behaviour policy.

---

> ### Author Response · Authors · 2021-11-19
> **Response to Reviewer #4**
>
> We thank the reviewer for their constructive feedback. We would like to address the following concerns:
>
> * *comparisons are still needed with other methods*. We thank the reviewer for the suggestion of comparing to GoExplore and a step-based policy weighting and adapting the external and intrinsic rewards; they will be great additions to our final paper or future work. We do compare with prominent research, such as with Burda et al., (2018). However, other related work uses different benchmarks, less well-suited for analyzing the *when* aspects of exploration, thus making direct comparisons fiddly.
>
> * *it is not clear to me how this* (fixing the strategy before the experiment) *helps the diversity argument of the method itself* – we thank the reviewer for the opportunity to clarify that, by selecting a priori an intra-episodic strategy, we do not force a mixture of explore and exploit modes within each episode; on the contrary, we give the agent full flexibility to explore at any granularity enumerated in Section 2.1 (be it step-level, experiment, episode, or within-episode), alternating between them and adapting from one strategy to another as learning progresses since the agent is powered by a (hyperparameter-free) meta-controller. To clarify, it is *the diversity of behaviour that is obtained within one variant of the mode-switching* that is the subject of our diversity claims and analysis (shown in Figures 3, 5 and 7) and **not** the *diversity across different granularities and switching mechanisms* as the reviewer questioned. Diversity is a desirable product of the intra-episodic exploration method applied, and it arises as the agent adapts the *when* answer throughout training. Under this framing, diverse behaviours are beneficial because they translate, by virtue of the fitness function in the meta-controller, to good performance too.

---

> > ### Comment · Reviewer_AkpW · 2021-11-20
> > **Follow-up**
> >
> > Thanks for your response.
> >
> > - Regarding Point 1, could you please clarify where the difficulty would be in comparing with such other methods? (In reference to this statement by the authors: "*However, other related work uses different benchmarks, less well-suited for analyzing the when aspects of exploration, thus making direct comparisons fiddly.*")
> >
> > - Point 2 clarifies my concern.
> >
> > - Could you also comment on this statement (from my review)? "*The authors point out that Montezuma’s Revenge and Phoenix have their best performance with different mode-switching behaviours, but these results have a really high variance. Perhaps, 3 seeds are just not enough and more seeds are needed to draw reliable conclusions.*"

---

> > > ### Author Response · Authors · 2021-11-22
> > > **Response to Follow-up**
> > >
> > > Thank you for following up on our response. We would like to clarify:
> > >
> > > * the difficulty in comparing with other related work (but *not* with GoExplore or with a step-based policy weighting, which we restate would make for excellent baselines we will try to follow up on) would come from the type of environments used in those related works. Our method is at its best in environments that exhibit a complex episodic structure and for which judicious switching would be required. We speculate this is less the case for, say, grid-worlds, so we stuck with Atari to ensure we place enough emphasis on the *when* aspects of exploration we were interested in.
> > >
> > > * When we show error bars, they are between the minimum and maximum values across our 3 seeds (not variance). Indeed, Montezuma’s Revenge and Phoenix are high variance games, and therefore, our best intra-episodic switching variants have also high min-max spans. Adding more seeds is always a good idea to sanity check results. However, there is a limit to how much we can run.

---

> > > > ### Comment · Reviewer_AkpW · 2021-11-22
> > > > **Thanks!**
> > > >
> > > > Thanks for the further clarifications. While I generally like the main idea and different investigations of this novel direction, I can't convince myself to fully advocate the acceptance of this paper. You propose a novel direction, and that's fantastic. But as the first paper in this direction, shouldn't the paper motivate the ideas a bit more on toy tasks first? Atari is great; but for an empirical paper, I really need to see justifications on simple illustrative tasks. Related to this, I partially side with the point of reviewer 3t1W and the response doesn't address this concern for me (Dabney et al. (2021) really do a nice job on experimentally motivating their exploration approach on simple problems, and then examine on Atari --- something along this line would make the paper very strong). For this reason, I keep my score (6).
> > > >
> > > > Will Dabney, Georg Ostrovski, André Barreto. Temporally-Extended ε-Greedy Exploration. ICLR 2021.

---

### Official Review · Reviewer_TbGe · 2021-11-01

**Correctness:** 4
**Technical Novelty And Significance:** 3
**Empirical Novelty And Significance:** 3
**Recommendation:** 8
**Confidence:** 3

**Details Of Ethics Concerns:**

Not applicable.

**Main Review:**

This paper conducts a series of experiments aimed at answering the question of when we should switch between exploitation and exploration during RL learning.
As positive points we can cite the wide related bibliography, from analogies with the animal system (human included) of exploring to other techniques such as the use of options. Another point to emphasize is the clear, careful and pleasurable writing that the authors developed in the paper. The authors managed to transmit a moment of reflection and expansion of thoughts about the possible behaviors shown by agents who learn with RL. Moments of reflection in an area that transforms at a very high speed are always welcome.

Perhaps the negative point is the limitation of the contribution -- since it is still a study that must be deepened in order to provide effective and efficient guidelines for RL system developers working in real applications.

However, it is an in-depth study, with interesting results. The article is worth being divulged to remind everyone working in the RL area that there is still a lot to study, investigate and evaluate in order to have robust, efficient and effective systems.

**Summary Of The Paper:**

This paper investigates when to switch between exploitation and exploration and how long to stay in each exploration mode during RL learning. It proposes new ways to explore the subject, especially with intra-episodic exploration variants. It presents a large body of study results (10 pages of appendices!!), and concludes with very thought-provoking suggestions and discussions.

**Summary Of The Review:**

The paper brings an in-depth study, with interesting results and very well written. It is worth being divulged to remind everyone who works in the RL area that there is still a lot to study, investigate and evaluate in order to have robust, efficient and effective systems.

---

> ### Author Response · Authors · 2021-11-19
> **Response to Reviewer #3**
>
> We thank the reviewer for the supportive and enthusiastic comments. We hope to continue to expand the breadth and depth of this research idea in future work.

---

### Official Review · Reviewer_ZCJR · 2021-11-02

**Correctness:** 4
**Technical Novelty And Significance:** 3
**Empirical Novelty And Significance:** 3
**Recommendation:** 8
**Confidence:** 4

**Main Review:**

Overall, the paper is very strong, well-motivated, and empirically sound. Most of my concerns are with clarifying details of the experiments and improving the exposition. These are listed below:

- paragraph 2 of the introduction: the typical answer to "when to explore" is when the agent is unfamiliar with the environment or its structure, i.e., early in its interactions with the environment. I would highlight why the heuristic of "explore earlier" should be challenged here instead.

- paragraph 3 of the introduction: it's not clear what the connection to schizophrenia here is, other than that this was a study of explore vs. exploit; are these individuals somehow impaired in choosing between these options, etc.?

- paragraph 1 of section 2 (methods): not sure that the example of riding a bicycle works - is the targeted acquisition of a new skill really exploration?

- section 2.1, description of "episode-level": it would be more useful to define "episode" in the context studied here, rather than using the example of training games vs. tournament matches.

- caption to Figure 1: the differences for D-G are unclear just by looking at this figure and the caption, other than that they are different intra-episode approaches; it would be better to clarify this

- section 2.2, description of "blind switching": this refers to "fractional episode length", but if we don't get to choose the length of an episode, how do we implement fractional episode length ex-ante?

- section 2.3, description of "bandit adaptation": the citations should be in-text citations

- section 3: it would be useful to explain the observed differences between these domains; are there hypotheses for why certain domains are better suited for different switching mechanisms? The discussion in Section 3.3 starts to get at this, but does not specifically address why the differences may arise.

- section 3.2: in appendix A.3, the compute budget is mentioned as being 2B frames; please clarify this difference

- section A.1, regarding using the full action sets: does this mean that some games have meaningless actions, i.e., actions that cause no effect in the world? how does this affect learning and why was this choice made?

- section A.1, regarding no life-loss signal: what exactly is an episode here if there is no life-loss signal? Is it just 108,000 frames (and why this number?) How does scoring work?

- section A.1: why was the choice to use raw, unprocessed frames made? in particular, what is the effect of keeping color information for learning?

- section A.2: is 1 TPU used (as mentioned here) or 2 TPUs (as mentioned in Section A.3)?

- section A.2: citations should be in-text citations; "," should be ";" after Schaul et al. (2021) in paragraph 2, line 4.

- section A.3: "while 2 TPUs" should be "using 2 TPUs".

- The captions to Figures 9 - 15, especially 9 and 10, should be expanded to clarify what the reader can learn from this figure (e.g., any hypotheses on what accounts for the differences between different environments)

**Summary Of The Paper:**

This paper studies switching between exploit and explore modes in reinforcement learning. It discusses switching mechanisms based on time ("blind switching") and based on state ("informed switching"). Studying seven Atari games, an empirical analysis of different switching mechanisms is performed.

**Summary Of The Review:**

The paper studies the relatively under-explored question of when agents should explore, introduces a novel exploration trigger called "value promise discrepancy", and performs a thorough empirical analysis in the domain of seven Atari games. Please see the main review for detailed comments and suggestions for improvement.

---

> ### Author Response · Authors · 2021-11-19
> **Response to Reviewer #2**
>
> We thank the reviewer for their careful reading of our paper and their appreciative comments. We have integrated their detailed suggestions in the newly updated version and they improved the clarity of the figures and explanations in our paper. We provide a few clarifications:
>
> * On the TPU usage: we use 1 single TPU for the learner, and 1 single TPU in batch inference, for aiding the CPU-based actors to gather experience, so in total 2 TPUs.
>
> * On the implementation of *blind* intra-episodic switching, we don't get to fix or pick the episode length (just as the reviewer points out too), so, in order to have access to this ex-ante, we keep an exponential moving average of the episodes encountered to calculate the fraction of steps to be spent in each mode.
> * Regarding some of the details of the Atari setup, while there is variation in the literature on the exact modalities, our approach is quite standard. Concretely, using raw frames without preprocessing is the simplest choice with the least domain-specific knowledge. Episodes end when the game is over, or after at most 108k frames (30 min in-game time), independently of how many in-game “lives” have been used up (which makes a difference in some games like Montezuma where sacrificing a life can grant access to later reward). Regarding the “full action set”, this  means that all games use the same (joystick-mimicking) 18-action interface, even if in some games that action set is redundant (e.g. “up” has the same effect as “up-and-fire”). For a more in-depth take on these questions, please see [1] and [2].  Note that for the purposes of our paper, other choices of these settings would likely give similar overall results, even if absolute scores would vary a bit.
>
>
> [1] "Revisiting the Arcade Learning Environment: Evaluation Protocols and Open Problems for General Agents" by Machado et al., 2018.
>
> [2] "On inductive biases in deep reinforcement learning" by Hessel et al., 2019.

---

> > ### Comment · Reviewer_ZCJR · 2021-11-20
> > **Re: Response to Reviewer #2**
> >
> > Thank you for your response. I've taken a look at the modified paper and remain in support of acceptance.

---

### Official Review · Reviewer_3t1W · 2021-11-03

**Correctness:** 2
**Technical Novelty And Significance:** 3
**Empirical Novelty And Significance:** 3
**Recommendation:** 6
**Confidence:** 3

**Main Review:**

Strengths:
- this paper investigates a novel area, which seems very important at a high level.
- it also proposes novel methods to start to address this area.

Weaknesses:
- the paper does not feel very focused. There are lots of different ideas and methods presented, but the takeaways are unclear.
- the figures are a bit hard to parse

First, I applaud the authors for tacking a new and unexplored area. However, the takeaways and next research steps are unclear. At least on Atari, the benefits of intra-episodic exploration seem limited.

Some suggestions for improving the paper:
- It would be helpful to have the different algorithm variants in Section 2 spelled out, especially regarding the different switching mechanisms. The current descriptions are quite high-level and it would be helpful to make them concrete.
- The tuple notation (shown in Figure 2) is a bit hard to parse. It would be helpful to show several examples, so that the differences in the different elements of the list can be seen more clearly.
- I think the paper would be a lot stronger if there were some tasks which more convincingly demonstrated the benefits of intra-episode exploration. This could be a new task which the authors design themselves. I think including the Atari experiments is useful in that it shows their methods can improve performance on a standard benchmark, however, the monolithic exploration methods already work quite well on these tasks, and it is not clear if there is that much more improvement to be had by exploring at a finer granularity. I do agree with the authors that in the “big picture”, monolithic exploration is likely suboptimal and more informed exploration will be necessary. I found their motivating example of learning to ride a bike while maintaining necessary daily activities to be very helpful. Can you design some task which distills this task in a simpler format? For example, some setup where the agent must regularly find food from a predictable source (exploiting) but also must explore when it can between finding food. Introducing new tasks which measure the ability to optimally switch between explore and exploit modes would also be useful to the community in building on this work.



**Summary Of The Paper:**

This paper proposes to study exploration at different levels of granularity. Current methods either explore at the level of individual steps (e.g., \epsilon-greedy), or at the level of experiments (e.g. first a reward-free exploration phase, followed by a task-dependent learning phase using the gathered data). This paper proposes to study exploration at the intra-episodic level, i.e. where the agent switches between exploration and exploitation within the same episode.

They discuss various design choices to perform exploration at this level, for example switching after a certain number of steps or with a certain probability, or switching based on the discrepancy between the predicted value and actual experienced value.

The experimental results show that including intra-episodic exploration gives a modest benefit over other exploration schemes when using an R2D2 base agent. Other insights are also included, which show that the proportion of exploration does change throughout the learning process, indicating that different degrees of exploration are useful at different stages. They also show that the informed switching component learns switching behaviors which are non-uniform throughout the episodes.

**Summary Of The Review:**

I'm on the fence about accepting this paper. On one hand I think it is good that the authors are exploring a new and important area and the ideas are interesting. On the other hand, this work still feels preliminary and the benefits of intra-episode exploration are not yet convincingly demonstrated.  I am not strongly opposed to accepting this paper since it could at least be a starting point for research in this area. However I think that if the authors could introduce new tasks where intra-episodic exploration convincingly helps, then I think this would be a very strong submission to a later conference.

---

> ### Author Response · Authors · 2021-11-19
> **Response to Reviewer #1**
>
> We thank the reviewer for recognizing and appreciating the novelty of our work. We also thank them for the many great suggestions and would like to respond to their following comments:
>
> *  *There are lots of different ideas and methods presented, but the takeaways are unclear* – we see two main takeaways (also discussed in Section 3.4, but they surely deserve more emphasis throughout the paper): (1) intra-episodic switching gives the flexibility needed to find the temporal sweet spot of exploration; and (2) it provides a fine-grained view which translates to rich, diverse, and beneficial behaviours.
>
> * *On Atari, the benefits of intra-episodic exploration seem limited*, with the reviewer suggesting to test on a *new task which the authors design themselves*. While we agree that this could indeed be *useful to the community in building on this work*, we think it is tricky to design such a task without over-specializing it for intra-episodic switching. That is why we avoided hand-crafting an environment and instead opted for a well-established RL task. Our goal was not to show an agent with intra-episodic switching achieving SOTA in *some* environment. Instead, we wanted to show our intra-episodic switching in an environment where our results can have credibility and where we can observe the undistorted strengths (and weaknesses) of this new method. Atari seemed to be the best testbed for the goals we set, and we offered some interesting qualitative results on it, complemented by some solid quantitative ones.
>
> *  *This work still feels preliminary and the benefits of intra-episode exploration are not yet convincingly demonstrated*. Indeed, we only scratched the surface of the intra-episodic switching exploration. However, we believe that we showed a new methodology, idea, and qualitative investigation that were interesting and sufficiently consistent in and of themselves. This paper is a starting point for research in this area, just as the reviewer so kindly acknowledges. We laid out the initial steps, and we hope to continue working on this promising, novel direction for exploration research.

---

### Author Response · Authors · 2021-11-19
**Response to all reviewers**

We thank all our reviewers for their detailed suggestions for improvements and their constructive comments. We have uploaded a modified version of our paper containing some (minor) clarifications and additions. We invite the reviewers to have a second look and add more questions and comments – we are willing to reiterate and refine the framing and presentation of our research ideas!

---

### Decision · Program_Chairs · 2022-01-20

**Decision:**

Accept (Spotlight)

**Comment:**

Exploration can happen at various levels of granularity and at different times during an episode,  and this work performs a study of the problem of exploration (when to explore/when to switch between exploring and exploitation, at what time-scale to do so, and what signals would be good triggers to switch). The study is performed on atari games.

Strenghts:
------------
The study is well motivated and the manuscript is overall well written
Studies a new problem area, and proposes an initial novel method for this problem
extensive study on atari problems

Weaknesses
--------------
some clarity issues as pointed out by the reviewers
no illustrative task is given to give a more intuitive exposition of the "when to explore" problem
comparison to some extra baselines like GoExplore would have been insightful

Rebuttal:
----------
Most clarity issues have been addressed satisfactorily. It has been explained why some requests for extra baselines would be challenging/or not relevant enough. While the authors agree that GoExplore would be an interesting baseline, they seem to have not added it. An illustrative task was not provided.

Summary:
------------
All reviewers agree that this manuscript opens up and tackles a novel direction in exploration, and provides an extensive empirical study on atari games (a standard benchmark for such problem settings). While I agree with the reviewers that point out that this paper could have been made stronger by adding an illustrative task and additional baselines like GoExplore, there is a general consensus that the provided empirical study on this novel problem setting is a good contribution in itself. Because of this I recommend accept.